# Stabilizing persistent currents in an atomtronic Josephson junction necklace

Luca Pezzè [1,2,3,8] ✉, Klejdja Xhani [1,2,3,8], Cyprien Daix [2,4,8], Nicola Grani[1,2,4], Beatrice Donelli [1,3,5], Francesco Scazza [1,2,6], Diego Hernandez-Rajkov[1,2], Woo Jin Kwon [7], Giulia Del Pace [2,4] & Giacomo Roati[1,2] ✉

Arrays of Josephson junctions are at the forefront of research on quantum circuitry for quantum computing, simulation, and metrology. They provide a testing bed for exploring a variety of fundamental physical effects where macroscopic phase coherence, nonlinearities, and dissipative mechanisms compete. Here we realize finite-circulation states in an atomtronic Josephson junction necklace, consisting of a tunable array of tunneling links in a ring-shaped superfluid. We study the stability diagram of the atomic flow by tuning both the circulation and the number of junctions. We predict theoretically and demonstrate experimentally that the atomic circuit withstands higher circulations (corresponding to higher critical currents) by increasing the number of Josephson links. The increased stability contrasts with the trend of the superfluid fraction – quantified by Leggett's criterion – which instead decreases with the number of junctions and the corresponding density depletion. Our results demonstrate atomic superfluids in mesoscopic structured ring potentials as excellent candidates for atomtronics applications, with prospects towards the observation of non-trivial macroscopic superpositions of current states.

Josephson junction arrays are pivotal and versatile elements that hold promise to turn quantum mechanics into emerging computing, sensing, and simulation technologies[1–6]. By harnessing the dissipationless non-linearity of single Josephson junctions, combined with strong collective effects, they show intriguing synchronization[7–10] and interference[11–13] phenomena. Furthermore, they serve as experimental tools to investigate the phase coherence and order parameters in high-$T_c$ superconductors[14,15].

An array of junctions in a multiply-connected geometry forms a Josephson junction necklace (JJN). In this configuration, the Josephson effect is used to control the current of persistent states, implementing robust dynamical regimes characterized by the competition between tunneling and interaction energies[16]. JJNs with one or two junctions realize common quantum interference devices (SQUIDs)[17,18], which find

applications in rotation sensing with superfluid gyroscopes[19,20] and magnetic field sensing with superconducting rings[17,21]. Furthermore, JJNs are key elements of atomtronic circuits[22–25]. Ultracold atoms in toroidal traps with a single junction or a weak link have been explored for the experimental realization of ultra-stable circulation states[26–29], including the study of various superfluid decay phenomena[30–32], current-phase relations[33] and hysteresis[34]. These experiments have stimulated several theoretical studies that have mainly focused on the analysis of different instability phenomena in ring superfluids with various types of defects and potentials[35–43]. In addition, double-junction atomtronic SQUIDs have enabled the observation of different regimes of Josephson dynamics[44], resistive flow[45] and quantum interference of currents[46]. Interestingly, as conjectured by Feynman[47], further intriguing quantum coherence effects can arise—due to the

[1]Istituto Nazionale di Ottica, Consiglio Nazionale delle Ricerche (CNR-INO), Largo Enrico Fermi 6, Firenze 50125, Italy. [2]European Laboratory for Nonlinear Spectroscopy (LENS), Via N. Carrara 1, Sesto Fiorentino 50019, Italy. [3]QSTAR, Largo Enrico Fermi 6, Firenze 50125, Italy. [4]Physics Department, University of Florence, Via Sansone 1, Sesto Fiorentino 50019, Italy. [5]University of Naples 'Federico II', Via Cinthia 21, Napoli 80126, Italy. [6]Physics Department, University of Trieste, Via A. Valerio 2, Trieste 34127, Italy. [7]Physics Department, Ulsan National Institute of Science and Technology (UNIST), Ulsan 44919, Republic of Korea. [8]These authors contributed equally: Luca Pezzè, Klejdja Xhani, Cyprien Daix. ✉e-mail: luca.pezze@ino.cnr.it; giacomo.roati@ino.cnr.it

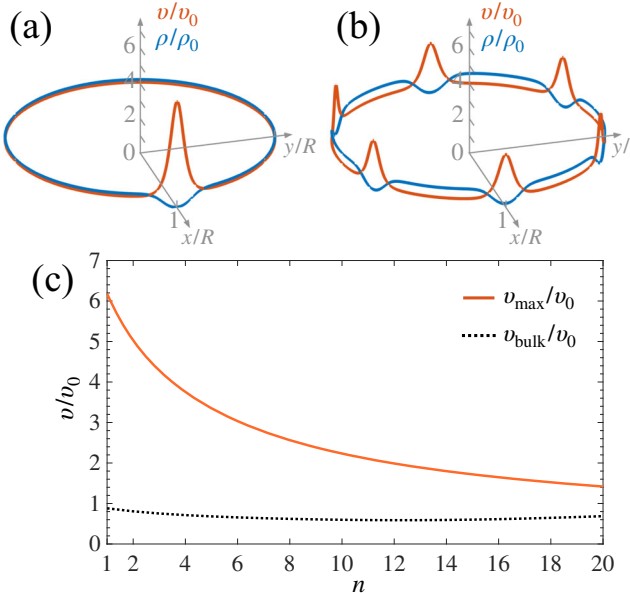

**Fig. 1 | Superfluid speed in a JJN. a, b** show the superfluid density $\rho$ (blue line) and speed $v$ (orange line) in a 1D JJN, divided by the values $\rho_0$ and $v_0$, respectively, of the homogeneous ring. The two panels correspond to $n=1$ (**a**) and $n=6$ (**b**) junctions. **c** Maximum, $v_{max}$ (solid orange line), and bulk, $v_{bulk}$ (dotted black line), superfluid speed as a function of the number of junctions. Results in all panels are obtained from the stationary states of the 1D GPE with $w=1$ and $\Omega=0$.

stiffness of the superfluid phase—in ring systems hosting arrays of multiple junctions. However, despite advancements both in manufacturing mesoscopic nanostructured multi-link circuits[48–52] and in engineering atomic trapping potentials[24,53–55], the realization of tunable JJNs with arbitrary number of junctions remains technologically and experimentally challenging, and so far elusive in both superconducting and superfluid platforms.

In this work, we investigate supercurrent states in an atomtronic JJN. We analytically predict the stabilization of persistent currents against decay by increasing the number of junctions, $n$. We support this surprising prediction by numerical simulations and we demonstrate it experimentally in a bosonic superfluid ring with $n$ up to 16. Such an effect is a direct consequence of the single-valuedness of the order parameter, reflecting the macroscopic phase coherence of the superfluid state. Increasing the number of Josephson links leads to a decrease of the superfluid speed across each junction and to the corresponding increase of the global maximum (critical) current in the ring. Furthermore, the density depletion associated to an increasing $n$ determines a decrease of the superfluid fraction according to Leggett's formulation[56,57] that, however, does not result in a decrease of the critical current. The full control of our atomtronic circuit opens exciting prospects toward the realization of non-trivial quantum superpositions of persistent currents[58–62].

## Results

### Critical current in a Josephson junction necklace

A steady superfluid state[63] can be described by a collective wavefunction $\psi(\boldsymbol{r})=|\psi(\boldsymbol{r})|e^{i\phi(\boldsymbol{r})}$. The phase $\phi(\boldsymbol{r})$ is related to the superfluid speed by $\boldsymbol{v}(\boldsymbol{r})=\frac{\hbar}{m}\nabla\phi(\boldsymbol{r})$, where $m$ is the particle mass and $\hbar$ the reduced Planck constant. To ensure a single-valued wavefunction, the integral of $\nabla\phi(\boldsymbol{r})$ calculated around any arbitrary closed path $C$ must be a multiple of $2\pi$:

$$\frac{m}{\hbar}\oint_C d\boldsymbol{r}\cdot\boldsymbol{v}(\boldsymbol{r})=2\pi w, \qquad (1)$$

where the integer (winding) number $w$ is a topological invariant. In a multiply-connected geometry (e.g., in a toroidal superfluid), Eq. (1) defines a series of quantized persistent-current states labeled by $w$[64,65]. Although the ground state is $w=0$, metastable finite-circulation states ($w\neq 0$) can be generated, as first demonstrated with liquid helium[66,67] and more recently with ultracold atomic gases[26,27,29,68–70].

In the following, we first illustrate the key ideas of this manuscript by studying the stationary states of the one-dimensional (1D) JJN. It consists of a ring of radius $R$ with $n$ equivalent junctions modeled as narrow Gaussian potential barriers and rotating with angular velocity $\Omega$. In the rotating frame, the current per particle is given by

$$J=\rho(\ell)[v(\ell)-\Omega R], \qquad (2)$$

where $\ell=R\theta$ is the coordinate along the ring, $\theta\in[0,2\pi]$ is the azimuthal angle and $\rho(\ell)$ is the superfluid density. For stationary states, we have $dJ/d\ell=0$ (continuity equation), which implies that $J$ is not only time-independent, but also spatially-constant (see Methods). As shown in Fig. 1a, b, Eq. (2) implies the interplay between density and speed: a dip of $\rho(\theta)=\rho(\ell)R$ [blue line, with $\rho(\theta)$ normalized to 1 and dimensionless], in correspondence with each barrier, is compensated by a local increase of $v(\theta)\equiv v(\ell)=\frac{\hbar}{mR}\frac{d\phi(\theta)}{d\theta}$ [orange line]. Here, $\rho(\theta)$ and $v(\theta)$ are calculated from the 1D Gross-Pitaevskii equation (GPE, see Methods). Comparing panels (a) and (b) of Fig. 1, obtained for the same value of the circulation $w$ and for different number of junctions, $n=1$ and $n=6$, respectively, we observe that the maximum superfluid speed, $v_{max}$, drops by increasing $n$. This is a consequence of the topological invariance expressed by Eq. (1). In fact, let us consider a JJN with $n$ equivalent junctions and write $v(\theta)=v_{bulk}+v_{n-peaks}(\theta)$, where $v_{bulk}$ is the bulk speed, given by the minimum velocity along the ring and $v_{n-peaks}(\theta)$ describes the $n$ peaks of the superfluid speed, see Fig. 1a, b. Replacing this expression for $v(\theta)$ into Eq. (1), we find

$$v_{bulk}+\frac{1}{2\pi}\int_0^{2\pi}d\theta\,v_{n-peaks}(\theta)=\frac{\hbar w}{mR}. \qquad (3)$$

The bulk contribution in Eq. (3) is expected to change only slightly when adding sufficiently-narrow junctions to the JJN [see the dotted black line in Fig. 1c]. On the contrary, the second term in Eq. (3) is proportional to $nv_{max}$. Therefore, for a given $w$, $v_{max}$ must decrease roughly as $1/n$ in order to keep the left-hand side of Eq. (3) constant. The decrease of $v_{max}$ is confirmed by the results of GPE simulations reported in Fig. 1c [solid orange line]. This effect directly implies a decrease of the phase gain across each junction, $\delta\phi=\frac{mR}{n\hbar}\int_0^{2\pi}d\theta\,v_{n-peaks}(\theta)$, upon increasing $n$. Using Eqs. (2) and (3), we find (see Methods for the detailed derivation)

$$\delta\phi=\frac{2\pi\tilde{w}}{n}\left(1-\frac{f(\tilde{w},n)}{2\pi\rho_{bulk}(\tilde{w},n)}\right), \qquad (4)$$

where $\rho_{bulk}(\tilde{w},n)$ is the bulk angular density, given by the maximum value of $\rho(\theta)$ along the ring, $\tilde{w}=w-\Omega/\Omega_R$ is an effective circulation in the rotating frame, and $\Omega_R=\hbar/(mR^2)$ is the rotational quantum[34]. Varying $\Omega$ allows to address continuous values of $\tilde{w}$. We also notice that $f(\tilde{w},n)\equiv(2\pi)^2\left[\int d\theta/\rho(\theta)\right]^{-1}\leq f_s$, where $f_s\in[0,1]$ is Leggett's superfluid fraction[56,57,71–74]. The latter expresses the phase rigidity of the system, quantified by the kinetic-energy response to a phase twist of the superfluid order parameter. In particular, $f(\tilde{w},n)=f_s$ for $w=0$ and in the limit $\Omega\to0$ (see Supplementary Information). In Fig. 2a, we plot $\delta\phi$ as a function of $n$, Eq. (4), where the quantities $f(\tilde{w},n)$ and $\rho_{bulk}(\tilde{w},n)$ are calculated from the stationary states of the GPE in the 1D ring. Symbols refer to different values of $\tilde{w}$. We clearly see that $\delta\phi$ decreases with $n$.

The decrease of $\delta\phi$ implies that the condition $\delta\phi_c\approx\pi/2$ – that determines the maximum (or critical) current $J_c$ in the JJN[15,75] – is met

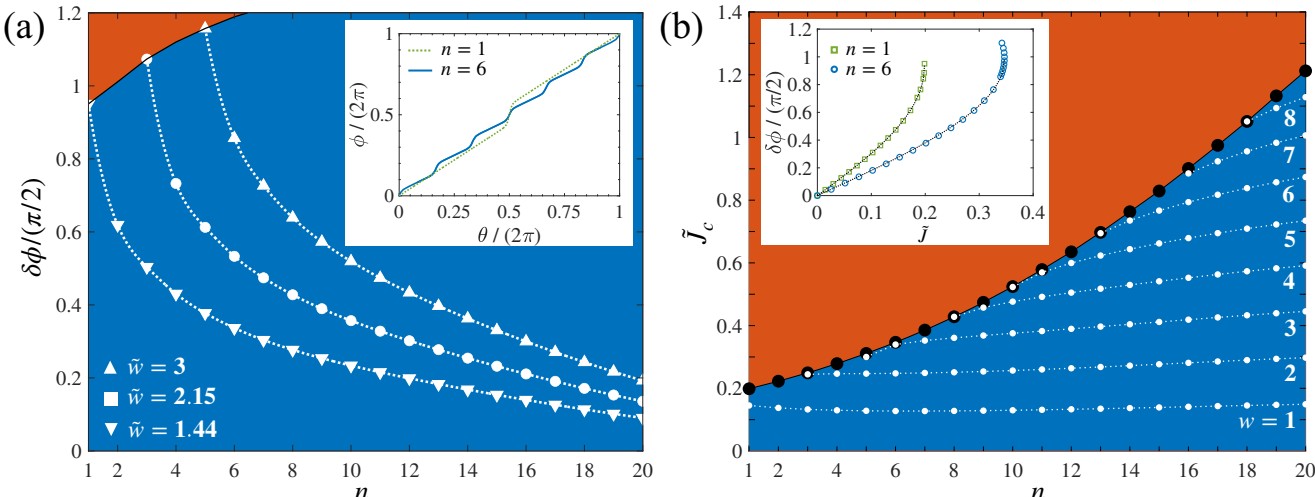

**Fig. 2 | Superfluid phase and critical current in a JJN. a** Phase gain $\delta\phi$ across each junction as a function of $n$, Eq. (4), where $f(\bar{w},n)$ and $\rho_{bulk}$ are obtained from GPE calculations in a 1D JJN. Symbols refer to $\bar{w} = 1.44$ (downward triangles), $\bar{w} = 2.15$ (squares) and $\bar{w} = 3$ (upward triangles). These correspond to the maximum values of $\bar{w}$, for $n = 1$, 3 and 5, respectively, for which a stable solution of the GPE can be found. For larger values of $\bar{w}$, the system is unstable, with the nucleation of solitons being observed in dynamical GPE simulation. Lines are guides to the eye. In particular, the solid black line connects maxima of $\delta\phi$ obtained for different $w$, separating the stable (blue) from the unstable (orange) region. The inset shows the superfluid phase $\phi$ as a function of the angle $\theta$ along the ring, for $n = 1$ (dotted green line) and $n = 6$ (solid blue line). **b** Critical current as a function of the number $n$ of junctions. The analytic formula, Eq. (5) (large black dots), reproduces the numerical calculation of the maximum current $\tilde{J}_c$. Small white dots show the current $\tilde{J}$ calculated for $\Omega = 0$ and different values of $w$, ranging from $w = 1$ (bottom) to $w = 8$ (top). Solid and dotted lines are guides to the eye. The orange region corresponds to values of the current above $\tilde{J}_c$ and are thus inaccessible in the system. Inset: $\delta\phi$ as a function of $\tilde{J}$ calculated for the stationary states of the 1D GPE, for $n = 1$ (green squares) and $n = 6$ (blue circles). The dotted lines are the current-phase relations $\delta\phi = \sin^{-1}(\tilde{J}/\tilde{J}_c) + 2\pi L\tilde{J}$[76] without free fitting parameters: the kinetic inductance $L$ is calculated from the relation $L = (\delta\phi_c - \pi/2)/(2\pi\tilde{J}_c)$, $\tilde{J}_c$ is the numerical maximum current and $\delta\phi_c$ is the corresponding value of the phase gain.

for higher values of $\bar{w}$ when increasing $n$. We find (see Methods)

$$\tilde{J}_c = \frac{nf_c/4}{2\pi[1 - f_c/(2\pi\rho_c)] + nf_cL},\tag{5}$$

where $\tilde{J}_c = J_c/\Omega_R$ is the rescaled critical current, $f_c$ and $\rho_c$ are the values of $f(\bar{w}, n)$ and $\rho_{bulk}(\bar{w}, n)$ obtained for $\tilde{J} = \tilde{J}_c$ (with $\tilde{J} = J/\Omega_R$), respectively. The dimensionless parameter $L$ in Eq. (5) is a small kinetic inductance associated to the finite width of the junction[20,30,76–78]. It is responsible for the deviation of $\delta\phi_c$ from $\pi/2$, as $\delta\phi_c = \pi/2 + 2\pi L\tilde{J}_c$[20,30,76–78]. From Eq. (5) it is apparent that the critical current is mainly determined by the competition between $n$ and $f_c$. In Fig. 2b we plot the $\tilde{J}_c$, obtained from the GPE solutions, as a function of $n$. Numerical values agree with Eq. (5) (black dots). Furthermore, small white dots in Fig. 2b show the current of metastable states in the case $\Omega = 0$ ($\bar{w} = w$), where $\tilde{J}$ assumes only quantized values (see Methods). The inset of Fig. 2b shows the numerical current-phase relation for $n = 1$ (green squares) and $n = 6$ (blue circles): results are well reproduced by $\delta\phi = \sin^{-1}(\tilde{J}/\tilde{J}_c) + 2\pi L\tilde{J}$[20,30,76–78] (dotted line). Figure. 2b and its inset clearly show that $\tilde{J}_c$ increases with the number of junctions. When $\tilde{J} > \tilde{J}_c$, the current enters the unstable regime [red regions in Fig. 2a, b], characterized, dynamically, by the simultaneous emission of $n$ solitons from the barriers (see refs. 40,41 for a study of the case $n = 1$).

Although the above discussion is restricted, for illustration sake, to a stationary 1D ring, the predicted effects are expected to hold qualitatively also in higher dimensional non-stationary systems in multiply-connected geometries, due to the general validity of Eq. (1). To confirm this expectation and mimic the experimental conditions, we have performed 3D time-dependent GPE simulations (see Methods). We prepare the ground state in the annular trap, impose a circulation $w_0$, and observe the dynamics of the system in the presence of $n$ junctions. Consistently with the results of Fig. 2, we observe a decrease with $n$ of both the superfluid speed and the time-averaged phase gain across each junction (see Supplementary Information). The

results of numerical simulations are schematically summarized as in Fig. 3a. If the number of junctions is below a critical value $n_c$ that depends on $w_0$, then vortices are emitted symmetrically from each barrier, causing phase slippage and a decay of the winding number in time (see Supplementary Information). This vortex emission is the 3D analogue of the observed simultaneous nucleation of $n$ solitons in 1D simulations in the unstable regime. If $n$ is increased above $n_c$, then the emission of vortices is suppressed and the circulation remains constant in time (upon neglecting thermally and quantum activated decay processes, which are estimated to occur on time scales larger than the experimental ones, see Supplementary Information). Our simulations show that a higher stable circulation corresponds to a larger time-averaged critical current.

## Experimental system and persistent current states

We investigate experimentally the predicted increase of current stability in JJNs by realizing a Bose-Einstein condensate (BEC) of $^6$Li molecules of mass $m = 2m_a$, where $m_a$ is the mass of a $^6$Li atom. The gas is held in an annular trap equipped with a variable number ($n \leq 16$) of static planar junctions. Both the ring-shaped trap and the array of junctions are produced by the same digital micromirror device (DMD) illuminated with blue-detuned light to provide a repulsive optical potential. Using the high resolution of the DMD projection setup, we create a dark ring-shaped region in the $x$-$y$ plane delimited by hard walls whose height is much larger than the chemical potential of the superfluid (given by $\mu/h \simeq 850$ Hz in the clean ring), with $R_{in} = 11.7 \pm 0.2$ μm and $R_{rout} = 20.6 \pm 0.2$ μm being the inner and outer radius of the annulus. The potential is completed by a tight harmonic confinement along the vertical $z$ direction, of trapping frequency $\omega_z = 2\pi \times (383 \pm 2)$ Hz. The junctions can be modeled as Gaussian peaks of initial height $V_0 \simeq (1.3 \pm 0.2)\mu$ and $1/e^2$-width $\sigma = (1.2 \pm 0.2)\xi$, with $\xi \approx 0.68$ μm being the healing length (see Supplementary Information for details on the barrier characterization). We initially trap approximately $6.8 \times 10^3$ condensed atom pairs inside the ring with

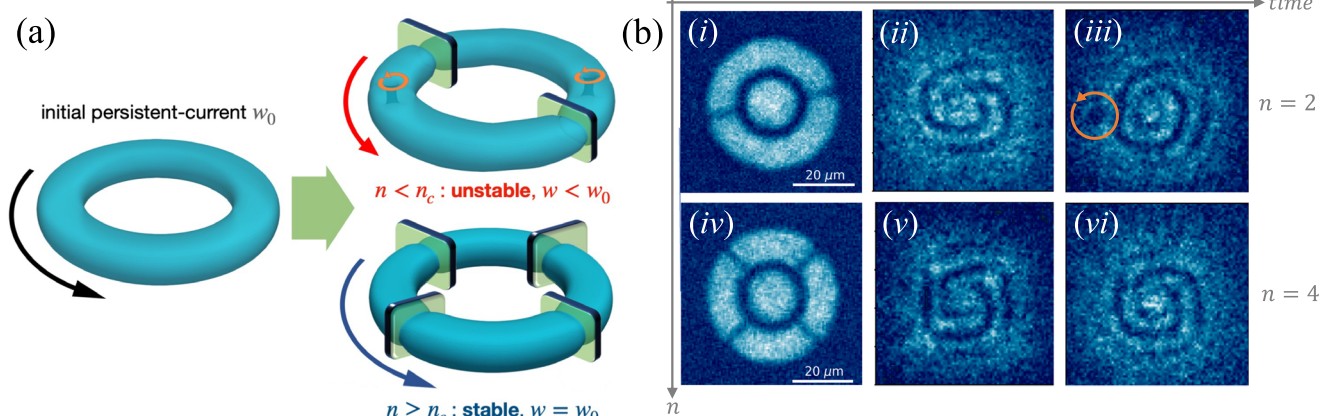

**Fig. 3 | Sketch of the experiment and observables. a** After preparing an initial persistent current state with circulation $w_0$, the $n$ junctions are ramped up (see text). The 3D density plots are isosurfaces obtained from 3D GPE numerical simulations of the experimental set-up. If $n$ is below a critical value $n_c$ depending on $w_0$, the initial current is dissipated via the nucleation of vortices (here $n = 2$ and vortices are highlighted by orange cycling arrows in the upper right plot). Conversely, if $n \geq n_c$ (here $n = 4$), the system remains stable with $w = w_0$ (lower right plot). **b** Examples of experimental in-situ images and interferograms obtained for $w_0 = 2$ and for the same number of junctions $n$ as in (**a**): $n = 2$ (unstable configuration), at $t = 0$ ($i$), $t = 1$ ms ($ii$) and $t = 7$ ms ($iii$); and $n = 4$ (stable configuration) for $t = 0$ ms ($iv$), $t = 1$ ms ($v$) and $t = 20$ ms ($vi$). In the case ($iii$), the circulation has decayed ($w(t) < w_0$) and the vortex emission is identified by the single spiral arm and the presence of a localized region of low density, i.e., a vortex.

a shot-to-shot stability around 5%. Due to the finite lifetime of our molecular BEC, the pair number decreases over the course of the current decay by at most 20%, causing a decrease of the chemical potential of the superfluid. Consequently the value of $V_0/\mu$ increases by up to ~15% depending on the holding time.

We initialize the superfluid ring in a quantized circulation state with winding number $w_0 \in \{1, 2, 3, 4\}$. Following the procedure described in ref. 29, different values of $w_0$ are obtained on-demand by shining a DMD-made azimuthal light intensity gradient onto the ring over a duration $t_I \ll \hbar/\mu$, i.e., shorter than the characteristic density response time, $\hbar/\mu$. In this way, we imprint a phase $\Phi(\theta) = U_0(\theta) \times t_I/\hbar$ to the condensate wavefunction without modifying the atomic density[70], where $U_0(\theta)$ is the spin-independent potential exerted by the light field on the atomic states that varies linearly with $\theta$[29]. After the imprinting, we wait 300 ms to let the cloud reach equilibrium, allowing the possible density excitations following the imprinting procedure to damp out[43]. We then progressively ramp up the $n$ Gaussian junctions over approximately 1 ms (corresponding to $\approx 6\,\hbar/\mu$). The barrier ramp-up time is adjusted to be slow with respect to the density response of the superfluid and fast relative to the typical current decay.

**Stability phase diagram**

To measure the winding $w$ in the ring, we exploit an interferometric probe[29,33,79]: we equip the atomic superfluid with a central disk acting as a phase reference [see panels (i) and (iv) in Fig. 3b] and measure the relative phase between the disk and the ring from the interference pattern arising after a short time-of-flight. The number of spiral arms in the interferogram provides access to the value of the circulation (winding number) at time $t$, $w(t)$. The different panels of Fig. 3b display typical examples of experimental images. In panels (i) and (iv) we show the in-situ atomic density profile at $t = 0$. The atomic density (averaged over 10 experimental images) is characterized by a homogeneous bulk both in the azimuthal and radial directions. The $n = 2$ (i) and $n = 4$ (iv) junctions are clearly visible and are associated to local dips in the density, similarly as in Fig. 1 and Fig. 3a. In panels (ii) and (iii) we show examples of spiral interference patterns emerging for an unstable dynamics, namely $w(t)$ decreasing in time below $w_0$ (here, $w_0 = 2$ and $n = 2$): in (ii) $t = 1$ ms and $w(t) = 2$, while in (iii) $t = 7$ ms and $w(t) = 1$. In particular, panel (iii) shows the presence of a vortex identified as a localized low-density defect and marked by the orange arrow. The vortex emission signals the decrease of $w$ by one quantum. In panels

(v) and (vi) we show instead the interferograms for stable dynamics, namely $w(t) = w_0$ (here, $w_0 = 2$ and $n = 4$). A non-circular, polygonal interference pattern is visible both at short [(v), $t = 1$ ms] and at long [(vi), $t = 20$ ms] times due to the sharp phase gain at the junctions.

By averaging the winding number over approximately 15 experimental realizations under the same conditions, we extract the evolution of the mean circulation $\langle w(t) \rangle$ for various $n$. We study the dynamics up to 250 ms, which is sufficient to observe steady current states at long-times while still limiting particle losses. The measured $\langle w(t) \rangle$ is shown in Fig. 4a for $w_0 = 2$. We fit each curve with an exponential decay given by $\langle w(t) \rangle = w_f + \Delta w \exp(-\Gamma t)$. The fitting parameters $w_f$, $\Delta w$ and $\Gamma$ allow us to characterize the mean supercurrent. As $\langle w(t) \rangle$ is obtained from statistical averaging, the figure shows that the number of realizations $w(t)$ that remain stable in time increases with the number of junctions. In particular, the number of stable realizations increases substantially when changing the number of junctions from $n = 2$ (red diamonds) to $n = 4$ (yellow squares). For $n = 10$ (blue circles), all realizations are stable: this demonstrates the experimental capability to create stable finite-circulation states in a JJN.

Figure 4b summarizes the results obtained for different $w_0$ and $n$, in the form of a stability phase diagram. In particular, we plot the quantity $\tilde{\Gamma} = \Delta w\, \Gamma\, /\, \max(\Delta w\, \Gamma)$, where each horizontal line of the phase diagram is normalized to its maximum value for fixed $w_0$. This quantity combines information on the difference between the initial and the final winding numbers, $\Delta w$, namely how much the currents decay, and on the timescale over which this decay takes place, $\Gamma$. Values of $\tilde{\Gamma} \approx 1$ (red regions) are obtained when most of the realizations $w(t)$ rapidly decay towards values of the circulation lower than the initial $w_0$. On the contrary, small values of $\tilde{\Gamma} \approx 0$ (blue regions) are obtained when most of the realizations are stable over time, namely $w(t) = w_0$. The phase diagram clearly shows that, on average, the system supports a higher number of stable realizations when increasing the number of junctions (see further details in Supplementary Information). The right axis of Fig. 4b reports the current of states with circulation $w_0$ in the clean ring. By the choice of normalization, $\tilde{\Gamma}$ shows a sharp transition from $\tilde{\Gamma} \approx 1$ to $\tilde{\Gamma} \approx 0$ when increasing $n$. The dashed white line in Fig. 4b denotes the critical winding number $w_c(n)$ and the corresponding current (right axes) as a function of $n$, as computed numerically from 3D GPE simulations. The numerical critical curve $w_c(n)$ is obtained for $V_0/\mu = 1.8$ and match the experimental phase diagram well. The need for a larger $V_0/\mu$ in numerical simulations with respect to the one

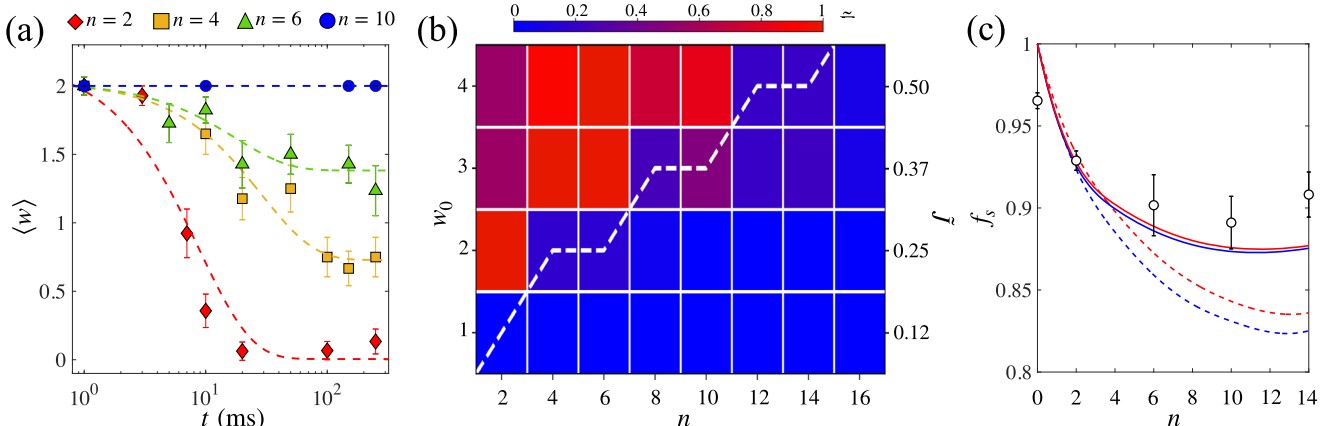

**Fig. 4 | Stability phase diagram of an atomtronic JJN. a** Mean circulation as a function of time, for $w_0 = 2$ and different number of barriers, $n$ (symbols), with averages and error bars obtained from 15 repeated measurements for each point. The dashed lines are exponential fits, $\langle w(t) \rangle = w_f + \Delta w \exp(-\Gamma t)$. **b** Effective decay rate $\bar{\Gamma} \propto \Delta w \Gamma$ (colormap), extracted from the exponential fits as in (**a**) as a function of $w_0$ and $n$. $\bar{\Gamma}$ quantifies the stability of an initial finite-circulation state $w_0$. The dashed white line is the critical circulation $w_c(n)$ as a function of $n$, obtained from 3D GPE simulations. The right axis shows the rescaled current $\bar{J}$ of $w_0$ states in the clean ring. **c** Upper (dashed red line) and lower (dashed blue line) bounds to the superfluid fraction $f_s$, Eq. (6), as a function of the number of junctions. Bounds are obtained from the ground state density of the numerical 3D GPE. The solid lines are the bounds evaluated by including the finite resolution of the experimental imaging system. Circles are the upper bound evaluated using experimental in-situ images and averaged over 10 realizations.

estimated in the experiment, is consistent with the finite lifetime of the sample (which implies that $V_0/\mu$ increases during the dynamics) and the finite resolution of the DMD potential, which makes the barriers not perfectly identical (see Supplementary Information). Anyway, we note that the only effect of a change of $V_0/\mu$ on the critical line $w_c(n)$ is to provide a linear shift, meaning that the particular choice of $V_0/\mu$ does not affect its trend, which well reproduce the experimental findings.

Given that $\tilde{J}_c(n) \sim n f_c$ from Eq. (5), a significant decrease of the superfluid fraction $f_s \geq f_c$ would overshadow the stabilization mechanism arising from increasing $n$. For this reason, in Fig. 4c, we study the dependence of $f_s$ on $n$ and indeed find a mildly decreasing trend, which is insufficient to disrupt the enhanced stability of currents for large $n$. According to a variational calculation by Leggett[56,57], the superfluid fraction $f_s$ can be bounded experimentally from the in-situ density profile $|\psi(\mathbf{r})|^2$ [72-74]:

$$\iint \frac{dz\,dr\,r}{\frac{1}{d^2}\int_{\text{cell}} \frac{d\theta}{|\psi(\mathbf{r})|^2}} \leq f_s \leq \left( \frac{1}{d^2} \int_{\text{cell}} \frac{d\theta}{\iint dz\,dr\,r\,|\psi(\mathbf{r})|^2} \right)^{-1}. \quad (6)$$

The bounds in Eq. (6) are computed by using the ground state of the 3D GPE (see Methods and Supplementary Information). We restrict the azimuthal angle $\theta$ over a unit cell of size $d = 2\pi/n$ and use the normalization $\iint dz\,dr\,r\int_{\text{cell}} d\theta\,|\psi(\mathbf{r})|^2 = 1$ [56,57,71]. In Fig. 4 we plot the upper (dashed red line) and lower (dashed blue line) bounds in Eq. (6). They are very close to each other as our system is approximately separable in the transverse spatial directions[72] and they coincide in 1D, where $f_s = \lim_{w=0, \Omega \to 0} f(\bar{w}, n)$ (see Supplementary Information). Increasing $n$ enhances the size of the density dip relative to the unit cell length and thus decreases both the lower and upper limits in Eq. (6), see Fig. 4c. Experimentally, for each value of $n$, we compute Leggett's upper bound on 10 different images of the experimental density. We compute the integral on the right-hand side of Eq. (6) by summing over all pixels inside an annular region with inner and outer radii $r_{cut1} > R_{in}$ and $r_{cut2} < R_{out}$ respectively. We have numerically verified that the values of the bounds do not depend on the exact size of this region. The corresponding mean values and standard deviations are shown as circles in Fig. 4a. The deviations from $f_s = 1$ in the clean ring ($n = 0$) are mainly due to noise in the experimental images, as well as the finite

pixel size of our imaging sensor. Experimental results are well reproduced when taking into account the finite resolution of the imaging system (solid blue and red lines) and show a slight decrease of $f_s$ with $n$.

## Discussion

Our work showcases the first experimental observation of ring supercurrents in periodic arrays of Josephson junctions. Such stable currents can be experimentally observed only for a sufficiently large number of links, as predicted by our theoretical model. In particular, our work shows that the maximum current flowing across the atomtronic circuit is due to a cooperative mechanism involving all the junctions rather than only to the properties of the single Josephson link. We expect the mechanism demonstrated in this manuscript to apply to any superfluids and superconductors as it solely depends on the single valuedness of the wavefunction in a multiply-connected topology.

Therefore, a natural extension of our work will be to investigate whether the same effect stabilizes supercurrents in other annular systems, such as atomic Fermi superfluids[28,29] and supersolids[80]. In the former case, the condensate fraction differs from unity even at $T = 0$ [81] and additional dissipative effects, such as Cooper pair-breaking[82,83] may compete with the stabilization mechanism. In the latter, intrinsic density modulations realize an array of self-induced Josephson junctions – as recently demonstrated in ref. 74 for an elongated atomic system – which can be controlled by tuning the confinement parameters.

Finally, the exquisite controllability offered by our platform opens the way toward realizing exotic quantum superposition of superflow states[58-62] with possible implications in both atomtronic and quantum technologies.

## Methods

### Derivation of Eqs. (4) and 5

Using Eq. (2) with the dimensionless $\rho(\theta) = \rho(\ell)R$, we obtain $v_{\text{bulk}} = JR/\rho_{\text{bulk}} + \Omega R$. We thus rewrite Eq. (3) as

$$\frac{2\pi \bar{J}}{\rho_{\text{bulk}}} + n\delta\phi = 2\pi\bar{w}. \quad (7)$$

Inserting Eq. (2) into Eq. (1) gives $\tilde{J}$ as a function of $\tilde{w}$ and $n$:

$$\tilde{J} = \frac{\tilde{w} f(\tilde{w}, n)}{2\pi}. \tag{8}$$

Finally, by replacing Eq. (8) into Eq. (7), we obtain Eq. (4).

To obtain Eq. (5), we notice that Eq. (4) is valid for every value of $\delta\phi$: in particular for the value $\delta\phi = \delta\phi_c$ achieved for $\tilde{J} = \tilde{J}_c$. Furthermore, $\delta\phi_c$ and $\tilde{J}_c$ are related as $\delta\phi_c = \pi/2 + 2\pi L\tilde{J}_c$, which follows from the current-phase relation $\delta\phi = \sin^{-1}(\tilde{J}/\tilde{J}_c) + 2\pi L\tilde{J}$[20,30,76–78]. Replacing this value into Eq. (4) and using Eq. (8), we find Eq. (5). The above current-phase relation models a Josephson junction with a finite width as a linear inductance in series with a purely sinusoidal one. Following ref. 76, we write $\delta\phi = \delta\phi_1 + \delta\phi_2$, where $\delta\phi_1$ is the phase drop across the sinusoidal inductance, namely $\tilde{J} = \tilde{J}_c \sin\delta\phi_1$, while $\delta\phi_2$ is the phase drop across the linear effective inductance, namely $\tilde{J} = \delta\phi_2/(2\pi L)$.

## Numerical methods
**1D GPE**. We consider the 1D GPE equation

$$i\hbar \frac{\partial}{\partial t} \psi(\ell, t) = \left[ -\frac{\hbar^2}{2m} \frac{\partial^2}{\partial \ell^2} + V(\ell) + g_{1D}|\psi(\ell, t)|^2 + i\hbar\Omega R \frac{\partial}{\partial \ell} \right] \psi(\ell, t), \tag{9}$$

where $\ell = R\theta$ is the spatial coordinate along the ring, $m$ is the molecule mass, $g_{1D}$ is an effective interaction parameter, $V(\theta) = V_0 \sum_{j=1}^{n} \exp[-2(\theta - \theta_j)^2/\sigma^2]$ is the necklace potential, given by the sum of Gaussian barriers centered at $\theta_j = 2\pi j/n$ and with amplitude $V_0$. We write $\psi(\ell, t) = \sqrt{\rho(\theta, t)} e^{i\phi(\theta, t)}/\sqrt{R}$ and search for stationary solutions of Eq. (9), namely $\frac{\partial\rho(\theta, t)}{\partial t} = 0$ and $\frac{\partial\phi(\theta, t)}{\partial t} = -\frac{\mu}{\hbar}$, where $\mu$ is the chemical potential. We obtain two coupled equations, corresponding to the real and imaginary part of Eq. (9), see e.g., ref. 84. The equation for the imaginary part is the continuity equation $\frac{\partial J(\theta)}{\partial \theta} = 0$. The equation for the real part writes as

$$\mu \sqrt{\rho(\theta)} = \left( -\frac{\hbar\Omega_R}{2} \frac{\partial^2}{\partial \theta^2} + \frac{\hbar\Omega_R}{2} \frac{\tilde{J}^2}{\rho(\theta)^2} + V(\theta) + \frac{g}{R}\rho(\theta) \right) \sqrt{\rho(\theta)}, \tag{10}$$

where we have used Eq. (2) to express the superfluid speed in terms of the current. Numerically, for a given value of $\tilde{w}$, we solve the two coupled Eqs. (8) and (10) iteratively. The free parameters $g$, $\sigma$ and $\tilde{V}_0$ are chosen in order to match the experimental conditions: $\sigma/\xi = 1.2$, $V_0/\mu_0 = 1.4$ and $\xi/R = 0.056$ (with $R = 12\,\mu m$ being approximately the inner radius of the experimental system), where $\mu_0$ is the chemical potential obtained in the homogeneous case (without barriers) and for $w = \Omega = 0$. Results of GPE simulations of the 1D JJN are reported in Figs. 1 and 2.

**3D GPE**. In order to better capture the experimental procedure and the dynamics of the system, in 3D we solve numerically the time-dependent GPE for static barriers,

$$i\hbar \frac{\partial \psi(\mathbf{r}, t)}{\partial t} = -\frac{\hbar^2}{2m} \nabla^2 \psi(\mathbf{r}, t) + V(\mathbf{r})\psi(\mathbf{r}, t) + g|\psi(\mathbf{r}, t)|^2 \psi(\mathbf{r}, t), \tag{11}$$

with $g = 4\pi\hbar^2 a/m$ the interaction strength, $a = 1010\,a_0$ the s-wave scattering length and $a_0$ the Bohr radius. The external trapping potential is $V(\mathbf{r}) = V_{harm}(\mathbf{r}) + V_{ring}(\mathbf{r}) + V_{barr}(\mathbf{r})$. Here, $V_{harm}(\mathbf{r}) = m(\omega_\perp^2 r^2 + \omega_z^2 z^2)/2$ is an harmonic confinement with $\{\omega_\perp, \omega_z\} = 2\pi \times \{2.5, 396\}$ Hz. The hard-wall potential creating the ring confinement in the $x$-$y$ plane is given by

$$V_{ring}(\mathbf{r}) = V_r \left[ \tanh\left( \frac{r - R_{out}}{d} \right) + 1 \right] + V_r \left[ \tanh\left( \frac{R_{in} - r}{d} \right) + 1 \right]. \tag{12}$$

with $R_{in} = 10.09\,\mu m$ and $R_{out} = 21.82\,\mu m$ being the inner and outer radius, respectively. The parameter $d = 1.1\,\mu m$ characterizes the

stiffness of the hard walls, fixed such that the numerical density profiles match the in-situ experimental ones. We take $V_r$ larger than the chemical potential $\mu$ such that the density goes to zero at the boundary. The $n$ barriers are modeled as identical Gaussian peaks of trapping potential

$$V_{barr} = V_0 \sum_{i=1}^{n/2} \exp\left[ -2(x\cos(i2\pi/n) + y\sin(i2\pi/n))^2/\sigma^2 \right]. \tag{13}$$

with constant width $\sigma = 0.8\,\mu m$. Notice that, similarly to the 1D case, taking $\psi(\mathbf{r}, t) = |\psi(\mathbf{r}, t)|e^{i\phi(\mathbf{r}, t)}$, Eq. (11) can be split in two coupled equations, one for its real and one for its imaginary part[84]. The continuity equation is $\frac{\partial}{\partial t}|\psi(\mathbf{r}, t)|^2 + \nabla \cdot \mathbf{j}(\mathbf{r}, t) = 0$, where $\mathbf{j}(\mathbf{r}, t) = -\frac{i\hbar}{2m}\left[ \psi^*(\mathbf{r}, t)\nabla\psi(\mathbf{r}, t) - \psi(\mathbf{r}, t)\nabla\psi^*(\mathbf{r}, t) \right]$ is the current density. The current per particle, $J$, is obtained by integrating $\mathbf{j}(\mathbf{r}, t)$ along a surface and it has thus the dimension of a frequency.

## Experimental methods
**Characterization of the tunneling barriers.** Due to the finite resolution of the DMD-projecting setup, the barriers of experimental JJNs are not identical. We characterize the properties of each barrier in the different configurations at various $n$ by acquiring an image of the DMD-created light profile by means of a secondary camera, and calibrating the optical potential via the equation of state of a BEC in a well characterized 3D harmonic trap[81]. Then, we extract the height and $1/e^2$-width by fitting the radially-averaged profile of each barrier with a Gaussian. From this set of data, we extract the mean values and standard deviation of barrier height $V_0 \simeq (1.3 \pm 0.2)\,\mu$ and width $\sigma = (1.2 \pm 0.2)\,\xi$. Error bars denote the standard deviation of the parameters over the set of barriers. Even though the barriers are not strictly identical, the obtained results show that it is possible to create similar barriers with fluctuations on $V_0$ and $\sigma$ that are only a fraction of the chemical potential and healing length, respectively.

**Imaging resolution.** To compare numerical and experimental data in Fig. 4c, we have taken into account the finite spatial resolution of the imaging system, characterized by a Point Spread Function (PSF) of full-witdh-half-maximum FWHM = $0.83\,\mu m$[81]. To estimate the theoretical curves of Fig. 4c, we first integrate the 3D numerical densities along the z direction, Then, we account for the finite experimental resolution by convolving the integrated numerical densities with a two-dimensional Gaussian with a FWHM matching the experimental PSF. This procedure leads to a decrease in the resolution of the density modulation, which causes the estimated superfluid fraction to increase and yields results in good agreement with experimentally extracted values, see Fig. 4c.

## Data availability
All the data supporting the findings of this study can be obtained from the corresponding authors on request.

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

## Acknowledgements

We thank Massimo Inguscio, Giovanni Modugno, Augusto Smerzi and Andrea Trombettoni for discussions. L.P. and K.X. acknowledge financial support by the QuantEra project SQUEIS. C.D. acknowledges funding from project MUR FARE TOPSPACE. F.S. acknowledges funding from the European Research Council (ERC) under the European Union's Horizon 2020 research and innovation programme (Grant agreement No. 949438) and from the Italian MUR under the FARE programme (project FastOrbit). G.R. acknowledges funding from the Italian Ministry of University and Research under the PRIN2017 project CEnTraL. G.R. and G.D.P. acknowledge financial support from the PNRR MUR project PE0000023-NQSTI. The Authors acknowledge support from the European Union - NextGenerationEU for the "Integrated Infrastructure initiative in Photonics and Quantum Sciences" - I-PHOQS [IR0000016, ID D2B8D520, CUP B53C22001750006]. This publication has received funding under Horizon Europe programme HORIZON-CL4-2022-QUANTUM-02-SGA via the project 101113690 (PASQuanS2.1)

## Author contributions

The theoretical work was carried out by L.P., K.X., and B.D. The measurements and the data analysis were carried out by C.D., N.G., F.S., D.H.-R., W.J.K, G.D.P, and G.R. All authors contributed to discuss the theoretical and experimental results and to write the manuscript.

## Competing interests

The authors declare no competing interests.
