## [Peer Review File · Nature Communications]

Stabilizing persistent currents in an atomtronic Josephson junction necklaceReviewer #1 (Remarks to the Author):

The manuscript presents impressive experimental results on persistent currents in Josephson junction arrays of ultracold atoms. The topic is timely in view of atomtronic applications, and only very few groups in the world have currently the possibility of performing this type of experiments. The results are really beautiful and the demonstration of robustness of the current in presence of several junctions is very elegant. The main result of the paper is sound and makes sense physically. However, surprisingly, the theory part of the manuscript is much weaker and even has some flaws in the equations. Therefore, I cannot recommend publication of the paper in the present form, but it would require substantial revisions of the theory part.

In detail, here are the main issues:

- 1) equation (2): in principle J is not strictly constant. One should *demonstrate it or illustrate it by numerical solution of the GPE
- 2) the '1DGPE' Eq.(9) used for the first part of the paper is not actually the usual GPE, it contains an effective term \tilde{J} which itself depends on the integrated density. The authors should use the microscopic 1D GPE in order that the argument fully stands alone - in the 'GPE' they use they already inserted some knowledge on the expected state. I acknowledge that they have presented 3D GPE results. But then Eq.(9) does not add much.
- 3) Equations (4),(5) and (6) are dimensionally wrong. The current and the frequency do not have the same dimensions. The 'current quantum' needs to be properly defined with the correct units, and the equations need to be corrected.
- 4) Overall I find the derivation of pages 2-3 too long and a bit confusing. The same equations are used several times in different ways without adding much. I suggest to reorganize the text in a more concise way, possibly moving derivation details or intermediate equations into an appendix.
- 5) Why the current-phase relation is imposed axiomatically (just before Eq. 7)? It must follow from some other equation. It does not seem to me compatible with Eq.(4), where there is no tunnel current. Similarly, the authors should derive or recall in an appendix the origin of the kinematic inductance in their equations.
- 6) The use of Leggett's formula is not very clear; the authors should physically explain the meaning of Leggett's expression as applied to their case. I also suggest to recall near Eq.(14) that it is not an exact expression but only an upper bound, stemming from a variational ansatz for the phase profile. The authors are actually in the position of verifying the Ansatz against 1D and 3D numerical simulations, I believe that this would increase the value of the paper.

Reviewer #2 (Remarks to the Author):

In this manuscript, the authors claim the observation of a persistent current in a Josephson junction necklace. The increase in stability with the increased number of junctions was explained theoretically, and the experimental observation confirmed this explanation. Finally, a superfluid fraction with density modulation was measured and compared with theoretical calculations. These results are noteworthy advancements from the previous works in atomtronics with Josephson junctions on a ring. This manuscript is the first step toward many exciting experiments to be investigated with a Josephson junction array on a ring. I recommend the acceptance of this manuscript to Nature Communications after the authors address the following questions.

Fig.7 shows that there is no qualitative change when the barrier height becomes lower than the chemical potential (weak link regime). It would be helpful to know if the experiments could be reproduced with weak links without tunneling junctions.

Regarding the decay of metastable current states, it might be good to have some rough estimate for the time scale of a thermal or quantum-activated decay process over the energy barrier. This might be

important for future experiments on observing superposition states between different current states.

The ramp up time of 1 ms was used for the barriers to be established. It will be informative to explain the reason for this time scale. Also it would be good to know if there were any experimental investigations on the variations of maximum current with different ramp up times.

RESPONSE TO THE REVIEWERS

We thank both Referees for the careful reading and for the useful and stimulating comments. Please find enclosed a point-by-point response and a list of corresponding changes in the manuscript. The constructive comments have certainly helped us to improve our manuscript.

In the response, we use color text to pinpoint the different text: in green we report the comments by the Referees, in black our response and in blue the corresponding changes in the manuscript.

In the main text, we highlight in blue the added text. In addition, we have revised Fig. 2(b), by replacing \tilde{J} with \tilde{J}_c in the vertical axis; Fig. 4(b), by aligning the ticks of the right axis for \tilde{J} with those of w_0 ; and Fig. 4(c), by reporting the results of a revised calculation for the solid and dashed lines (changes with respect to the previous lines are minimal).

Reviewer #1:

The manuscript presents impressive experimental results on persistent currents in Josephson junction arrays of ultracold atoms. The topic is timely in view of atomtronic applications, and only very few groups in the world have currently the possibility of performing this type of experiments. The results are really beautiful and the demonstration of robustness of the current in presence of several junctions is very elegant. The main result of the paper is sound and makes sense physically. However, surprisingly, the theory part of the manuscript is much weaker and even has some flaws in the equations. Therefore, I cannot recommend publication of the paper in the present form, but it would require substantial revisions of the theory part.

We thank the Referee for the careful reading of our manuscript and for the favorable evaluation of our work. In response to the Referee's suggestions, we have extensively revised the presentation of the theory part of the manuscript. Enclosed, please find a comprehensive response to each comment along with a list detailing the principal changes made.

In detail, here are the main issues:

1) equation (2): in principle J is not strictly constant. One should demonstrate it or illustrate it by numerical solution of the GPE.

Let us elaborate under which conditions the current J is both spatially and temporally constant. Let us start from the continuity equation

$$\frac{\partial |\psi(\mathbf{r}, t)|^2}{\partial t} + \nabla \cdot \mathbf{j}(\mathbf{r}, t) = 0, \quad (\text{S1})$$

where

$$\mathbf{j}(\mathbf{r}, t) = -\frac{i\hbar}{2m} (\psi^*(\mathbf{r}, t) \nabla \psi(\mathbf{r}, t) - \psi(\mathbf{r}, t) \nabla \psi^*(\mathbf{r}, t)), \quad (\text{S2})$$

is the current density, m is the particle mass and $\psi(\mathbf{r}, t)$ is the superfluid order parameter. For stationary states, we have $\frac{\partial |\psi(\mathbf{r}, t)|^2}{\partial t} = 0$ and thus the current density has a null divergence:

$$\nabla \cdot \mathbf{j}(\mathbf{r}) = 0. \quad (\text{S3})$$

In the initial part of the manuscript [around Eq. (2)], we study stationary states and thus use Eq. (S3). For these states the current is time-independent and, following Eq. (S3) and the Gauss's law, we have that the current $J = \int_{\Sigma} d\mathbf{S} \cdot \mathbf{j}(r)$, where $d\mathbf{S}$ is the differential cross-sectional area vector, is also spatially-constant along the ring.

In 1D, the current and the current density coincide, $j = J$, and we can thus find a simple expression for J , Eq. (2). This is a significant simplification enabling us to derive equations that analytically show the reduction in both the maximum superfluid speed and the phase gain across each junction as the number of junctions, n , increases, while the critical current concurrently rises. This aspect was undoubtedly insufficiently clarified in the prior version of the manuscript. We have now added the following sentences:

Lines 98-100: "In the following, we first illustrate the key ideas of this manuscript by studying the stationary states of the one-dimensional (1D) JJN."

Lines 106-109: “For stationary states, we have $dJ/d\theta = 0$ (continuity equation), which implies that J is not only time-independent, but also spatially-constant along the ring (see Methods). As shown in Fig. 1(a-b), Eq. (2) implies the interplay between density and speed: a dip of $\rho(\theta)$ [blue line], in the correspondence of each barrier, is compensated by a local increase of $v(\theta)$ [orange line]. Here, $\rho(\theta)$ and $v(\theta)$ are calculated from the 1D Gross-Pitaevskii equation (GPE, see Methods).”

While we focus the first part of the manuscript on studying the stationary states in a 1D ring, the predicted phenomena hold qualitatively also for non-stationary superfluid states in higher dimensions. In fact, the single valuedness requirement for the wave function, which holds in any multiply-connected geometry, implies that the maximum superfluid speed decreases with n : $v_{\max} \propto w/n$. Since the phase gain at each junction, $\delta\phi$, is proportional to v_{\max} , it immediately follows that $\delta\phi \propto w/n$ as well. Finally, in a JJN, it is expected that the maximum current is observed for $\delta\phi_c \sim \pi/2$, which is reached for larger values of w when n is increased. Larger values of w also imply a larger critical current.

These qualitative expectations are confirmed by our full 3D numerical GPE simulations as well as in the experiment. We clarify that the 3D simulations follow the experimental methods and consider a phase imprinting: in the 3D numerics, we do not address stationary states and, for winding numbers smaller than the critical one, the current slightly oscillates in time around a time-independent mean value. Nevertheless, we observe the decrease of maximum speed, the decrease of $\delta\phi$ as well as the increase of critical current with the number of junctions, see Fig. 5(b), (c) and (d), respectively, where time-averaging is considered.

To clarify the above aspects we have added the following sentences:

Lines 175-182: “Although the above discussion is restricted, for illustration sake, to a stationary 1D ring, the predicted effects are expected to hold qualitatively also in higher dimensional non-stationary systems in multiply-connected geometries, due to the general validity of Eq. (1). To confirm this expectation and mimic the experimental conditions, we have performed 3D time-dependent Gross-Pitaevskii simulations (see Methods).”

Lines 196-197: “Our simulations show that a higher stable circulation corresponds to a larger time-averaged critical current.”

2) the ‘1DGPE’ Eq.(9) used for the first part of the paper is not actually the usual GPE, it contains an effective term \tilde{J} which itself depends on the integrated density. The authors should use the microscopic 1D GPE in order that the argument fully stands alone - in the ‘GPE’ they use they already inserted some knowledge on the expected state. I acknowledge that they have presented 3D GPE results. But then Eq.(9) does not add much.

We thank the Referee for her/his comment: this gives us the opportunity to provide more details regarding the derivation of Eq. (9) [corresponding to Eq. (10) in the new version of the manuscript]. Let us show that Eq. (10) *does coincide* with the usual GPE. We follow and adapt here the methods that can be found in the classic book by Pethick and Smiths [83]. Let us start with the standard form of the 1D GPE in a ring trap of radius R rotating at angular velocity Ω . In the rotating frame, we have

$$i\hbar \frac{\partial}{\partial t} \psi(x, t) = \left[-\frac{\hbar^2}{2m} \frac{\partial^2}{\partial x^2} + V(x) + g_{1D} |\psi(x, t)|^2 + i\hbar \Omega R \frac{\partial}{\partial x} \right] \psi(x, t), \quad (\text{S11})$$

where $x = R\theta$ is the spatial coordinate along the circumference, $\theta \in [0, 2\pi]$ is the angle along the ring, m is the molecule mass, g_{1D} is an effective interaction parameter, $V(x)$ is the necklace potential and $\psi(x, t)$ is normalized according to $\int_0^{2\pi R} dx |\psi(x, t)|^2 = 1$, with the integral extending on the full circumference of the radius R . We further consider $\psi(x, t) = \sqrt{\rho(\theta, t)} e^{i\phi(\theta, t)} / \sqrt{R}$, where $\phi(\theta, t)$ is the superfluid phase,

$$v(\theta, t) = \frac{\hbar}{mR} \frac{\partial \phi(\theta, t)}{\partial \theta} \quad (\text{S13})$$

is the superfluid speed [$v(\theta, t) - \Omega R$ is the speed in the rotating frame] and $\rho(\theta, t)$ is normalized according to $\int_0^{2\pi} d\theta \rho(\theta, t) = 1$. Equation (S11) can thus be split in two coupled equations: one for the real and one for the imaginary part. The equation for the imaginary part is

$$\frac{\partial \rho(\theta, t)}{\partial t} = -\frac{\partial J(\theta, t)}{\partial \theta}, \quad (\text{S14})$$

where

$$J(\theta, t) = \rho(\theta, t)(v(\theta, t)/R - \Omega) \quad (\text{S15})$$

is the superfluid current density in the rotating frame coinciding, in 1D, to the current. Notice that J has the dimension of 1/sec, the same as Ω (below, we will further elaborate on this point). The equation for the real part is

$$-\hbar \sqrt{\rho(\theta, t)} \frac{\partial \phi(\theta, t)}{\partial t} = \left(-\frac{\hbar^2}{2mR^2} \frac{\partial^2}{\partial \theta^2} + \frac{m}{2} v(\theta, t) + V(\theta) + \frac{g}{R} \rho(\theta, t) - mR\Omega v(\theta, t) \right) \sqrt{\rho(\theta, t)}. \quad (\text{S16})$$

In the following (as well as in the manuscript), we look for *stationary solutions* of the above coupled equations, namely

$$\frac{\partial \rho(\theta, t)}{\partial t} = 0, \quad (\text{S17a})$$

$$\frac{\partial \phi(\theta, t)}{\partial t} = -\frac{\mu}{\hbar}, \quad (\text{S17b})$$

where μ is the chemical potential. As mentioned above, the continuity equation (S14) with the stationary requirement Eq. (S17a) implies that J is constant at every point along the ring as well as in time. We write

$$J = \rho(\theta)(v(\theta)/R - \Omega). \quad (\text{S18})$$

This equation implies

$$\frac{v(\theta)}{R} = \frac{J}{\rho(\theta)} + \Omega. \quad (\text{S19})$$

We insert Eq. (S19) into the quantization condition in the ring,

$$\frac{mR}{\hbar} \int_0^{2\pi} d\theta v(\theta) = 2\pi w, \quad (\text{S20})$$

giving

$$\frac{mR^2}{\hbar} \int_0^{2\pi} d\theta \left(\frac{J}{\rho(\theta)} + \Omega \right) = 2\pi w. \quad (\text{S21})$$

Recalling that J is spatially-independent, we can bring it out of the integral and find

$$\frac{J}{\Omega_R} = \frac{2\pi(w - \Omega/\Omega_R)}{\int_0^{2\pi} d\theta \frac{1}{\rho(\theta)}}, \quad (\text{S22})$$

where $\Omega_R = \hbar/(mR^2)$ (called ‘‘rotation quantum’’ in the literature, see Ref. [35]) has the dimensions of 1/sec and sets the natural unit for the current. In the rescaled units used in the manuscript ($\tilde{J} = J/\Omega_R$ and $\tilde{w} = w - \Omega/\Omega_R$), Eq. (S22) rewrites as

$$\tilde{J} = \frac{2\pi\tilde{w}}{\int_0^{2\pi} d\theta \frac{1}{\rho(\theta)}}, \quad (\text{S23})$$

Using the stationary condition for the phase, see Eq. (S17b), and the expression for the superfluid speed in terms of the current, Eq. (S19), Eq. (S16) writes as

$$\mu \sqrt{\rho(\theta)} = \left(-\frac{\hbar^2}{2mR^2} \frac{\partial^2}{\partial \theta^2} + \frac{\hbar^2}{2mR^2} \frac{\tilde{J}^2}{\rho(\theta)^2} + V(\theta) + \frac{g}{R} \rho(\theta) \right) \sqrt{\rho(\theta)}. \quad (\text{S24})$$

This equation coincide with Eq. (9) of the previous version of the manuscript. Numerically, we set the quantity $\tilde{w} = w - \Omega/\Omega_R$ and solve the two coupled equations (S23) and (S24) iteratively, using a Runge-Kutta method and starting from a trial density function. At each imaginary time evolution step, *i*) $\rho(\theta)$ is updated according to Eq. (S24); *ii*) it is replaced into Eq. (S23) to calculate the current \tilde{J} ; and *iii*) the updated current is finally replaced into Eq. (S24). The procedure is repeated iteratively. In practice, we have chosen the above coupled-equation method because it proves to be numerically very stable and directly includes the single-valuedness condition of the wavefunction into Eq. (S23).

We have summarized the above discussion in the Methods section ‘‘1D GPE’’. This subsection now includes Eq. (S11) [as Eq. 9] and Eq. (S24) [as Eq. 10].

In the section 3D GPE, we have added the following sentences (Lines 441-446): “Notice that, similarly to the 1D case, taking $\psi(\mathbf{r}, t) = |\psi(\mathbf{r}, t)|e^{i\phi(\mathbf{r}, t)}$, Eq. (11) can be split in two coupled equations, one for its real and one for its imaginary part [83]. The continuity equation is $\frac{\partial}{\partial t}|\psi(\mathbf{r}, t)|^2 + \nabla \cdot \mathbf{j}(\mathbf{r}, t) = 0$, where $\mathbf{j}(\mathbf{r}, t) = -\frac{i\hbar}{2m}[\psi^*(\mathbf{r}, t)\nabla\psi(\mathbf{r}, t) - \psi(\mathbf{r}, t)\nabla\psi^*(\mathbf{r}, t)]$ is the current density.”

3) Equations (4),(5) and (6) are dimensionally wrong. The current and the frequency do not have the same dimensions. The ‘current quantum’ needs to be properly defined with the correct units, and the equations needs to be corrected.

We thank the Referee for highlighting a point of possible confusion, which we have now mended. In Eq. (2) of the main text, J is the current per particle in 1D. Since, in 1D the current coincides with the current density and it is a scalar (as mentioned above), Eq. (2) is consistent with Eq. (S2). More generally, the current density \mathbf{j} has the dimensions $1/(\text{meters}^{D-1}\text{sec})$, where D is the spatial dimension. This can be seen by noticing that the order parameter $\psi(\mathbf{r}, t)$ must satisfy the normalization condition $\int d\mathbf{r} |\psi(\mathbf{r}, t)|^2 = 1$ and it has therefore the dimensions of $1/\text{meter}^{D/2}$. The current is obtained by integrating the density current on a section of the system: therefore, in any D , the current has dimensions of $1/\text{sec}$.

As clarified in the new version of the manuscript, the angular density $\rho(\theta)$ is dimensionless. Therefore the quantities ρ_{bulk} and $f \equiv (2\pi)^2 [\int d\theta/\rho(\theta)]^{-1}$ are dimensionless as well. We find that Eq. (4) in the new version of the manuscript [Eq. (6) in the old version] is consistent with $\delta\phi$ being a (dimensionless) phase. Equation (4) and (5) in the old version of the manuscript are now Eqs. (7) and (8), respectively. There, \tilde{J} is the (dimensionless) rescaled current. To summarize, we believe that the equations pointed out by the Referee are dimensionally correct as they involve, on both sides, dimensionless quantities.

The natural units for the current is the quantity $\hbar/(mR^2)$: in fact, in the 1D ring without junctions, Eq. (8) gives $J = \frac{w}{2\pi} \frac{\hbar}{mR^2}$ (notice that $f(\tilde{w}, n = 0) = 1$ in the clean ring). Consistently with the current (see above), the quantity $\hbar/(mR^2)$ has the dimensions of $1/\text{sec}$ and was indicated as “current quantum” in the previous version of the manuscript. Following the literature, see for instance Ref. [33], we now prefer to call it the “rotational quantum” and indicate it with the symbol Ω_R that was used in other works on ultracold atoms in ring traps.

Finally notice that we use the symbol J (or \tilde{J}) in 3D to indicate the current, consistent with the 1D notation. In our simulations, J obtained by integrating the current density along a surface area.

We have added the following sentences to clarify the above aspects in the manuscript:

Line 103. Before Eq. (2) we clarify that J refers to “the current per particle”.

Lines 104-106. To clarify the dimensions of the different quantities in Eq. (2) we have added “ $\rho(\theta)$ is the dimensionless angular density normalized to one and $v(\theta) = \frac{\hbar}{mR} \frac{d\phi(\theta)}{d\theta}$ is the superfluid (angular) speed.”.

The quantity $\Omega_R = \hbar/(mR^2)$ is defined at line 137-138, including the reference [33]: “ $\Omega_R = \hbar/(mR^2)$ is the rotational quantum [33]”.

We have added (line 446-449): “The current per particle, J , is obtained by integrating $\mathbf{j}(\mathbf{r}, t)$ along a surface area and it has thus the dimension of a frequency: in 1D the current coincides with the current density.”.

We have clarified that the symbol m in manuscript refers to the mass of a particle in the superfluid gas (namely, a Li molecule), which is twice the mass of a ${}^6\text{Li}$ atom (line 200-202): “a Bose-Einstein condensate (BEC) of ${}^6\text{Li}$ molecules of mass $m = 2m_a$, where m_a is the mass of a ${}^6\text{Li}$ atom.”.

4) Overall I find the derivation of pages 2-3 too long and a bit confusing. The same equations are used several times in different ways without adding much. I suggest to reorganize the text in a more concise way, possibly moving derivation details or intermediate equations into an appendix.

We have followed the Referee’s suggestion and have simplified the discussion on pages 2 and 3. The text flows as follows:

- i) We rewrite the condition for the single valuedness of the wavefunction, Eq. (1), in 1D and in terms of the superfluid bulk and peak speed, obtaining Eq. (3). This allows to argue that the maximum superfluid speed, v_{max} decreases with n . This is confirmed by numerical results in Fig. 1(c).
- ii) The decrease of v_{max} implies that the phase $\delta\phi$ across each junction decreases with n as well, see Eq. (4) and Fig. 2(a). Details on the derivation of Eq. (4) are now reported in the Methods.

iii) The decrease of $\delta\phi$ implies the increase of the critical current, Eq. (5). This is confirmed in Fig. 2(b). Details on the derivation of Eq. (5) are reported in the Methods.

We thank the Referee for stimulating this major rearrangement of the main text that certainly allowed us to improve the section on pages 2-3.

The derivation of Eqs. (4) and (5) is now reported in a new subsection in the Methods, page 6. The text around Eqs. (4) and (5) has been adjusted accordingly.

5) Why the current-phase relation is imposed axiomatically (just before Eq. 7)? It must follow from some other equation. It (the current-phase relation) does not seem to me compatible with Eq.(4), where there is no tunnel current. Similarly, the authors should derive or recall in an appendix the origin of the kinematic inductance in their equations.

We thank the Referee for raising this question. We clarify that an analytical relation between the current J and the phase gain $\delta\phi$ across each junction is necessary to obtain Eq. (5) – namely the analytical expression for the critical current as a function of n . To this aim, we use the current-phase relation first introduced by Deaver and Pierce for superconductors [75]. These authors consider a Josephson junction with a finite width and assume that it consists of a linear inductance in series with a purely sinusoidal one. They write $\delta\phi = \delta\phi_1 + \delta\phi_2$, where $\delta\phi_1$ is the phase drop across the sinusoidal inductance, namely $J = J_c \sin \delta\phi_1$, while $\delta\phi_2$ is the phase drop across the linear effective inductance, namely $J = \delta\phi_2/(2\pi\ell)$. Combining the equation we obtain [75]

$$\delta\phi = \sin^{-1}(J/J_c) + 2\pi\ell J. \quad (\text{S26})$$

In the context of superfluids, the Deaver-Pierce current-phase relation has been used to discuss experimental findings with ^3He [76] and ^4He [77] (see also a discussion in the review [19]) as well as Bose-Einstein condensates [29].

We clarify that, in the numerics, we do not impose any current-phase relation. The value of $\delta\phi$ and J are obtained numerically from the stationary states of the 1DGPE (see discussion above). We find that the current phase-relation Eq. (S26) reproduces very well the numerical results *without any fitting parameter*: J_c in Eq. (S26) is given by the maximum current, while the inductance ℓ is obtained from the relation $\ell = (\delta\phi_c - \pi/2)/(2\pi J_c)$, where $\delta\phi_c$ is the phase gain across each junction calculated at $J = J_c$. This is now clarified in the main text. Notice that Eq. (S26) also well reproduces the current-phase numerical data in 3D, see Fig. 5(d).

All these important aspects are now clarified in the new version of the manuscript:

Lines 156-159: “The dimensionless parameter ℓ in Eq. (5) is a small kinetic inductance associated to the finite width of the junction [19, 29, 75–77]. It is responsible for the deviation of $\delta\phi_c$ from $\pi/2$, as $\delta\phi_c = \pi/2 + 2\pi\ell\tilde{J}_c$ [19, 29, 75–77].”.

Lines 165-169: “The inset of Fig. 2(b) shows the current-phase relation for $n = 1$ (green squares) and $n = 6$ (blue circles): results are well reproduced by $\delta\phi = \sin^{-1}(\tilde{J}/\tilde{J}_c) - 2\pi\ell\tilde{J}$ [19, 29, 75–77] (dotted line).”.

Last lines in the caption of Fig. 2b: “The dotted lines are the current-phase relation $\delta\phi = \arcsin(\tilde{J}/\tilde{J}_c) - 2\pi\ell\tilde{J}$ [76] without free fitting parameters: the kinetic inductance ℓ is calculated from the relation $\ell = (\delta\phi_c - \pi/2)/(2\pi\tilde{J}_c)$, \tilde{J}_c is the numerical maximum current and $\delta\phi_c$ is the corresponding value of the phase gain.”.

Lines 401-411: “Furthermore, $\delta\phi_c$ and J_c are related as $\delta\phi_c = \pi/2 - 2\pi\ell\tilde{J}_c$, which follows from the current-phase relation $\delta\phi = \sin^{-1}(\tilde{J}/\tilde{J}_c) - 2\pi\ell\tilde{J}$ [19, 29, 75–77]. Replacing this value into Eq. (4) and using Eq. (8), we find Eq. (5). The above current-phase relation models a Josephson junction with a finite width as a linear inductance in series with a purely sinusoidal one. Following Ref. [75], we write $\delta\phi = \delta\phi_1 + \delta\phi_2$, where $\delta\phi_1$ is the phase drop across the sinusoidal inductance, namely $J = J_c \sin \delta\phi_1$, while $\delta\phi_2$ is the phase drop across the linear effective inductance, namely $J = \delta\phi_2/(2\pi\ell)$.”

6) The use of Leggett’s formula is not very clear; the authors should physically explain the meaning of Leggett’s expression as applied to their case. I also suggest to recall near Eq.(14) that it is not an exact expression but only an upper bound,stemming from a variational ansatz for the phase profile. The authors are actually in the position of verifying the ansatz against 1D and 3D numerical simulations, I believe that this would increase the value of the paper.

We thank the Referee for the very interesting suggestion. We recall that Leggett’s formula is given by Eq. (1) in Ref. [56]:

$$f_s = - \lim_{|\Omega| \rightarrow 0} \frac{L_z}{I_{cl}\Omega}, \quad (\text{S27})$$

where $L = \int d\mathbf{r} m |\psi(\mathbf{r})|^2 \mathbf{r} \times \mathbf{v}(\mathbf{r})$ is the expectation value the angular momentum and $I_{cl} = \int d\mathbf{r} m r^2 |\psi(\mathbf{r})|^2$ is the classical moment of inertia.

In a 1D ring of radius R , the computation of Eq. (S27) simplifies substantially: we have $I_{cl} = mR^2$ and $L_z = mR \int d\theta \rho(\theta) v(\theta)$, where $\rho(\theta)$ is the dimensionless density along the ring (see above). Using Eq. (S19), we thus find

$$f_s = 1 - 2\pi \lim_{\Omega \rightarrow 0} \frac{J/\Omega_R}{\Omega/\Omega_R} \quad (\text{S28})$$

Replacing into this equation the expression for J/Ω_R given in Eq. (S22), we find

$$f_s = \frac{1}{\frac{1}{(2\pi)^2} \int_0^{2\pi} d\theta \frac{1}{\rho(\theta)}}, \quad (\text{S29})$$

where the superfluid density $\rho(\theta)$ is calculated for the stationary state with $w = 0$ and $\Omega = 0$. For a more direct comparison with Ref. [56], we restrict the integration to a unit cell of size d (corresponding to the angular distance between two peaks) and normalize $\rho(\theta)$ to 1 within that cell. In this case, Eq. (S29) rewrites as [Eq. (14) in the previous version of the manuscript, Eq. (15) in the revised version]

$$f_s = \frac{1}{\frac{1}{d^2} \int_{\text{cell}} d\theta \frac{1}{\rho(\theta)}}. \quad (\text{S30})$$

We emphasize that Eq. (S30) is not an upper bound but the actual exact expression for f_s . In fact, it is possible to see that in 1D Leggett's upper and lower bounds coincide.

In the Methods section, we have clarified that Eq. (15) coincide with the exact expression for Leggett's superfluid density in 1D. The above discussion has been added in the last section of the Methods.

In 3D, the Referee's suggestion to numerically compute Eq. (S27) and compare it to Leggett's upper and lower bounds indeed raises an interesting and relevant point, especially given the active research in evaluating superfluid fractions in various ultracold atoms platforms and geometries, as referenced in the manuscript. However, it is acknowledged that a comprehensive computation of f_s for the system described in the manuscript is a substantial and intricate research endeavor that extends well beyond the manuscript's scope. Two potential complementary approaches for calculating f_s using Eq. (S27) are identified:

1. computing the stationary states of the 3D JJN in a rotating frame at angular velocity Ω ;
2. employing dynamical simulations with rotating junctions, which would mimic a realistic experimental procedures.

Stimulated by the Referee's comment, we anticipate (based on preliminary numerical results) that Eq. (S27) can be influenced by finite-size effects due to the finite width of the ring. Rotating a thick ring does not merely introduce an azimuthal phase twist to the superfluid, as assumed by Leggett in the limit of large radius and small width, but it also induces transversal effects that potentially increase the value of f_s compared to Leggett's variational estimate. These considerations highlight the complexities involved in accurately computing f_s for the described system and suggest avenues for further investigations. We emphasize that we computed Leggett's bounds to confirm the smooth decrease of the superfluid fraction with the number of junction: this remains true also in the presence of finite size effects and supports our results regarding the increase of the critical current.

Lines 650-655: to address the comment by the Referee, we have added the following sentences: "It should be noticed that the variational bounds in Eq. (6) have been obtained by Leggett [55,56] by considering a small phase twist in a narrow ring of relatively large radius. In our system, the superfluid fraction evaluated according to Eq. (14) may include additional transversal finite-size effects due to the non-negligible thickness of the ring."

Reviewer #2:

In this manuscript, the authors claim the observation of a persistent current in a Josephson junction necklace. The increase in stability with the increased number of junctions was explained theoretically, and the experimental observation confirmed this explanation. Finally, a superfluid fraction with density modulation was measured and compared with theoretical calculations. These results are noteworthy advancements from the previous works in atomtronics with Josephson junctions on a ring. This manuscript is the first step toward many exciting experiments to be investigated with a Josephson junction array on a ring. I recommend the acceptance of this manuscript to Nature Communications after the authors address the following questions.

We thank the Referee for the very positive assessment of our work. Enclosed, you will find responses to all comments and questions.

1) Fig.7 shows that there is no qualitative change when the barrier height becomes lower than the chemical potential (weak link regime). It would be helpful to know if the experiments could be reproduced with weak links without tunneling junctions.

We clarify that the observed effect is theoretically predicted to hold also in the weak-link regime. We find a smooth crossover from the Josephson to the weak-link regime, where the main difference is that the circulation for which the system becomes unstable, e.g. in the case of a single junction $n = 1$, increases while decreasing the ratio V_0/μ . This trend is clearly observed in the numerics shown in Fig. 7(a). We agree with the Referee that it would be very interesting to extend our experimental exploration also in the weak-link regime. However, this requires to increase the fidelity of the preparation of high circulation ($w_0 \geq 5$) states, as outlined in Ref. [28]. Thus, the investigation into the weak-link regime is left as a future task.

Lines 523-528: we have added the following sentences: “We finally notice that the increase of $w_c(n)$ with n shows no qualitative change when the barrier height becomes lower than the chemical potential (weak link regime). However, clearly observing the transition from unstable to stable current states as a function of n for $V_0 < \mu$ would require the preparation of initial states with circulations much larger than those considered in this work.”.

2) Regarding the decay of metastable current states, it might be good to have some rough estimate for the time scale of a thermal or quantum-activated decay process over the energy barrier. This might be important for future experiments on observing superposition states between different current states.

Following the advice of the Referee, we estimate the thermal and quantum-activated decay rate for the metastable state corresponding to $w = 1$. The system free energy is described by a washboard-like potential where different metastable states, corresponding to the energy minima, are separated by an energy barrier. In particular, for the states $w = 0$ and $w = 1$ this energy barrier E_b is estimated as $E_b = 2E_J$, where E_J is the Josephson energy, proportional to the critical current. Following the Refs. [85,86] we estimate the thermal-activated decay rate $\nu_{th} = \nu_J \exp(-2E_J/k_B T)$, where $\nu_J = \sqrt{E_J E_c}/\hbar$ is the plasma frequency with E_c being the interaction energy. This expression gives a value $\nu_{th} \approx 0.54\text{Hz}$ for the experimental temperature $T = 60\text{nK}$, i.e. a phase slippage rate of 1.84 s. Furthermore, we also investigate the effect of temperature on the decay of supercurrents for winding number values $w_0 = w_c$. This study was performed at the experimental temperature by making use of a self-consistent finite temperature model, known as collisionless Zaremba-Nikuni-Griffin model. We found that the current remains persistent for at least 150ms (no phase slippage) which is the time evolution we choose for all our 3D simulations. Regarding quantum effects, using the tunneling rate $\nu_q \approx \nu_J \exp(-2E_J/\hbar\nu_J)$ [85,86], we estimate a decay time around 150 s. Interestingly, the increase of E_J (and thus E_b) with the number of barriers n results in a decrease in both the quantum and thermal phase slippage rates. Increasing the number of junctions thus makes the system less prone to phase-slippage processes.

We have added the new paragraph “*Rate of thermally- and quantum-activated decay processes*” in the Methods section, where we summarize the above discussion. It reads (lines 554-570): “The energy barrier separating the $w = 0$ from the $w = \pm 1$ states can be estimated as twice the Josephson energy (see e.g. [29], linearly dependent to the critical current. Following Refs. [84,85], we estimate a thermally-activated decay rate $\nu_{th} \approx 0.54\text{ Hz}$ for the experimental temperature $T = 60\text{ nK}$ (at which the condensate fraction is 80%), i.e., a phase slippage time of 1.84 s. We have checked numerically – by solving the collisionless Zaremba-Nikuni-Griffin model [86,87] at the experimental temperature – that finite-temperature dissipation does not affect the critical winding number ($w_0 = w_c$) for at least 150 ms and marginally affects the decay time. Furthermore, by following Ref. [84], we estimate a typical quantum tunnelling-induced decay time of the order of hundreds of seconds. Note that both thermal and quantum phase slippage times further increase with the critical current and therefore with the number n of junctions.”

3) The ramp up time of 1 ms was used for the barriers to be established. It will be informative to explain the reason for this time scale. Also it would be good to know if there were any experimental investigations on the variations of maximum current with different ramp up times.

We thank the Referee for this question. In principle, we would like the barriers to be raised slowly and in an adiabatic manner. However, there is also an upper limit on the ramp up time set by the timescale of the fastest decay rates we observed which were on the order of a few ms, see e.g. figure 9 pannel (c). The final ramp up time of 1 ms was chosen as an acceptable compromise with respect to those requirements.

Lines 239-242: we have added the sentence “The barrier ramp-up time is adjusted to be slow with respect to the density response of the superfluid and fast relative to the typical current decay [see Fig. 4(a)].”

Reviewer #1 (Remarks to the Author):

The authors have substantially revised the manuscript. The new added parts answer my questions and make the manuscript of much better readability, and of increased quality. I am now in favour of publication of the manuscript.

There are a couple of minor points that would be a pity not to correct

1) in one dimension the current and the integrated current are not the same: in one dimension, one can of course define a current density $j(\theta)$ as well as the integrated current $J = \int d\theta j(\theta)$. The hydrodynamic equations hold for $j(\theta, t)$, so in a general out-of-equilibrium setting one should clearly make the difference among the two. In the stationary state the two coincide as follows from the continuity equation. But it is wrong to say that in 1D $J = j(\theta)$ always.

2) I find weird the definition of the 'dimensionless density $\rho(\theta)$ '. In fact Equation(2) is fully general for a **dimensionful** density $\rho(x)$: $j(x) = \rho(x) (v(x) - \Omega R)$. I suggest the authors to write Eq.(2) in its most general, dimensionful way, and introduced reduced dimensions just below Eq.(2).

Reviewer #2 (Remarks to the Author):

The authors addressed the questions and comments satisfactorily, and I recommend the publication of the manuscript.

Reply to Referee A

We thank the Referee for recommending the publication of our manuscript. Please, find enclosed a reply to the final points raised by the Referee:

- There are a couple of minor points that would be a pity not to correct

1) in one dimension the current and the integrated current are not the same: in one dimension, one can of course define a current density $j(\theta)$ as well as the integrated current $J = \int d\theta j(\theta)$. The hydrodynamic equations hold for $j(\theta, t)$, so in a general out-of-equilibrium setting one should clearly make the difference among the two. In the stationary state the two coincide as follows from the continuity equation. But it is wrong to say that in 1D $J = j(\theta)$ always.

We have removed the text claiming $J = j(\theta)$ in 1D.

- 2) I find weird the definition of the 'dimensionless density $\rho(\theta)$ '. In fact Equation(2) is fully general for a **dimensionful** density $\rho(x)$: $j(x) = \rho(x) (v(x) - \Omega R)$. I suggest the authors to write Eq.(2) in its most general, dimensionful way, and introduced reduced dimensions just below Eq.(2).

We have revised Eq. (2) following the Referee's suggestion. The text around that equation has been adjusted accordingly.